# Regional Prevalence of Intermediate *Leptospira* spp. in Humans: A Meta-Analysis

**DOI:** 10.3390/pathogens10080943

**Published:** 2021-07-27

**Authors:** Aina Nadheera Abd Rahman, Nurul Husna Hasnul Hadi, Zhong Sun, Karuppiah Thilakavathy, Narcisse Joseph

**Affiliations:** 1Department of Biomedical Science, Faculty of Medicine and Health Sciences, Universiti Putra Malaysia, Serdang 43400, Selangor, Malaysia; 192388@student.upm.edu.my (A.N.A.R.); 192182@student.upm.edu.my (N.H.H.H.); cifer03@gmail.com (Z.S.); 2Genetics and Regenerative Medicine Research Group, Faculty of Medicine and Health Sciences, Universiti Putra Malaysia, Serdang 43400, Selangor, Malaysia; 3Department of Medical Microbiology, Faculty of Medicine and Health Sciences, Universiti Putra Malaysia, Serdang 43400, Selangor, Malaysia

**Keywords:** intermediate *Leptospira*, human leptospirosis, prevalence, meta-analysis

## Abstract

Leptospirosis is one of the most widespread bacterial diseases caused by pathogenic *Leptospira*. There are broad clinical manifestations due to varied pathogenicity of *Leptospira* spp., which can be classified into three clusters such as pathogenic, intermediate, and saprophytic. Intermediate *Leptospira* spp. can either be pathogenic or non-pathogenic and they have been reported to cause mild to severe forms of leptospirosis in several studies, contributing to the disease burden. Hence, this study aimed to estimate the global prevalence of intermediate *Leptospira* spp. in humans using meta-analysis with region-wise stratification. The articles were searched from three databases which include PubMed, Scopus, and ScienceDirect. Seven studies were included consisting of two regions based on United Nations geo-scheme regions, among 469 records identified. Statistical analysis was performed using RevMan software. The overall prevalence estimate of intermediate *Leptospira* spp. in humans was 86% and the pooled prevalences were 96% and 17% for the American and Asia regions, respectively. The data also revealed that *Leptospira wolffii* was the most predominantly found compared to the other intermediate species identified from the included studies, which were *Leptospira inadai* and *Leptospira broomii*. The estimated prevalence data from this study could be used to develop better control and intervention strategies in combating human leptospirosis.

## 1. Introduction

Leptospirosis is one of the most well-known, widespread zoonotic diseases, which accounts for high morbidity and mortality particularly in the regions with humid tropical or subtropical climates and in areas with impoverished populations. Leptospirosis has been estimated to affect around 1.03 million people and causes 58,900 deaths every year [1]. Even though the reported data is significant, there is no precise estimation of the global burden of human leptospirosis as it is often overlooked due to the wide range of clinical manifestations such as fever, diarrhea, headache, vomiting, muscle aches, malaise, jaundice, renal failure, pulmonary hemorrhage, etc. [2]. Leptospirosis also mimics several other diseases, for instance, dengue fever, malaria infection, influenza infection, Hanta virus, viral flu-like illnesses, and typhoid fever [3]. Moreover, leptospirosis is usually under-reported because of the poor health surveillance especially in the under-developed and developing countries. Therefore, leptospirosis was reported as one of the bacterial neglected tropical diseases due to its high disease burden and huge impacts on the public health following the re-emergence of this disease in several parts of the world [4].

This disease is caused by the pathogenic spirochetes of the genus *Leptospira*. *Leptospira* can be mainly classified according to methods used that are either based on a serological classification system or molecular classification system [5]. Traditionally, serology-based methods identified leptospires according to their antigenic properties found on the outer membrane of the bacteria, which was associated with the structural heterogeneity of lipopolysaccharides (LPS) [6]. This method divides these bacteria into two: *Leptospira interrogans* and *Leptospira biflexa*, which contain pathogenic and non-pathogenic strains, respectively. There are 26 serogroups and more than 300 serovars *Leptospira* currently identified using this classification system, which were usually detected by agglutination techniques such as Microscopic Agglutination Test (MAT) and cross agglutination absorption test (CAAT) [7,8]. On the other hand, phylogenetic or genomic classification system based on DNA relatedness using DNA-DNA hybridization and 16S-rRNA-based methods have further categorized 22 Leptospira species into three different clusters, which comprised of pathogenic, intermediate, and saprophytic. There are currently 10 pathogenic *Leptospira* spp. (*Leptospira noguchii*, *Leptospira kirschneri*, *Leptospira interrogans*, *Leptospira santarosai*, *Leptospira mayottensis*, *Leptospira borgpetersenii*, *Leptospira alexanderi*, *Leptospira weilii*, *Leptospira alstonii*, and *Leptospira kmetyi*) that could cause the disease. Meanwhile, there are five intermediate *Leptospira* spp. (*Leptospira broomii*, *Leptospira inadai*, *Leptospira fainei*, *Leptospira wolffii*, and *Leptospira licerasiae*) that have uncertain pathogenicity but mostly cause moderate symptoms, and seven saprophytic *Leptospira* spp. (*Leptospira meyeri*, *Leptospira wolbachii*, *Leptospira terpstrae*, *Leptospira vanthielii*, *Leptospira biflexa*, *Leptospira yanagawae*, and *Leptospira idonii*) that are commonly found in water and soil and unable to infect people [9]. Although serological classification using MAT technique remains the gold standard method, the overall prediction of the infecting species is not as reliable as genomic classification. This is because molecular methods allow the identification of the species exhibiting both pathogenic and non-pathogenic serovars, the intermediate *Leptospira* spp. 

Intermediate *Leptospira* spp. were reported to cause mild to severe forms of leptospirosis in humans, however, the findings about its pathogenicity status are still unclear [10]. This indicates the need for more studies on the intermediate species since they were also isolated from the clinical samples and proved to have virulence features, which have the potential to (or might) affect the burden of leptospirosis [11]. In addition, there have been many individual studies and research that successfully identified the presence of intermediate *Leptospira* spp. in various areas, countries, or regions. However, to date, there is no reported meta-analysis of published data that summarize its prevalence in humans on a global scale. Therefore, it is essential to conduct meta-analysis to systematically summarize the relevant individual studies in a similar field and to obtain a more precise estimation on the overall effect measure [12]. 

This study aimed to estimate the overall prevalence of intermediate *Leptospira* spp. in humans by quantitatively synthesizing the frequency of its presence in humans from different regions via meta-analysis. 

## 2. Materials and Methods

### 2.1. Literature Search Strategy

Meta-analysis for this study was conducted according to the PRISMA (Preferred Reporting Items for Systematic Reviews and Meta-Analyses) 2009 guidelines [13]. Comprehensive search related to the human leptospirosis caused by the intermediate *Leptospira* spp. was performed using three databases, which include PubMed, ScienceDirect, and Scopus. The combinations of the keywords “prevalence”, “presence”, “epidemiology”, “leptospirosis”, “intermediate Leptospira”, “human”, “patient”, and species of the intermediate *Leptospira* were included for the search terms for the relevant studies (Table 1). Boolean connectors such as “OR” and “AND” were applied to connect the terms within and between the categories, respectively. In addition, truncation and wildcard operators such as ‘*’, ‘#’, or ‘$’ were also utilized to maximize the search for the related terms of the pertinent studies. The search strategy was slightly adjusted based on the requirements of different databases. There was no restriction posed for publication dates and languages during the initial search. The reference lists of the included studies were searched manually to seek for additional relevant papers that were not selected during the initial search. The last database search was carried out on 10 January 2021.

### 2.2. Eligibility Criteria and Study Selection

Inclusion and exclusion criteria were pre-determined using PICOS (population, intervention, comparator, outcome, study design) approach [13] (Table 2). The subjects of the studies were any individuals suspected with leptospirosis infection including those that were co-infected with the other diseases. There was no restriction imposed on age, gender, or race of the subjects. The studies were excluded if there was no leptospirosis infection and if the studies did not report the origin of the samples or patients. Any irrelevant study was also removed. As for the types of intervention, the studies must identify the presence of intermediate *Leptospira* spp., which were detected using any recognized diagnostic or confirmation methods. The species of the intermediate *Leptospira* spp. and the methods used must be specified. The outcome measures were the frequency of the samples positive for intermediate *Leptospira* spp. and the total number of positive samples investigated. These data were used to determine the raw prevalence outcome, which were measured in percentage (%) by dividing the number of positive samples with the total number of positive samples tested in the study. All research articles of any countries and publication year were included. However, inaccessible full-text articles, letter to editor, duplicated publications, studies using other than English language, and secondary research such as review paper, systematic reviews, and meta-analyses were removed.

All the search results from the three databases were compiled and sorted out in one Microsoft Excel spreadsheet document. The articles with redundant titles and authors were removed. The title and abstract of the remaining studies were screened by three reviewers, independently according to the inclusion and exclusion criteria. The results of the screening by the three reviewers were cross-checked, and disagreements were resolved through consensus. The qualified studies were then subjected to full-text screening to further ascertain their relevancy and studies that did not meet the criteria were excluded. The flow of study selection was done by referring to the PRISMA flow diagram [13].

### 2.3. Data Extraction

Data from the selected studies were extracted independently by two reviewers, which included the first author and year of publication, location where the samples were collected, number of samples positive for intermediate *Leptospira* spp., number of samples positive for pathogenic *Leptospira* spp., total sample size confirmed for leptospirosis, method used for the confirmation of leptospirosis, and the type of intermediate *Leptospira* spp. Data collected from both of the reviewers were compared and re-checked for better accuracy. Disagreements and inconsistencies in data extraction were resolved through consensus and discussion with the third reviewer (Z.S.). 

### 2.4. Quality Assessment

Quality assessment of the included studies was conducted by two reviewers independently in accordance with the modified Critical Appraisal Checklist recommended by the Joanna Briggs Institute (JBI) since the use of this tool had been formally evaluated and increased in assessing prevalence studies [14,15]. Any disagreements were discussed and resolved through consensus with the third reviewer. The tool was comprised of nine questions, by which the reviewers referred to when assessing the studies. The score is either 0 or 1 for the answers of “No/Unclear” and “Yes”, respectively. Therefore, the score can range from 0 to 9. 

### 2.5. Statistical Analysis

The extracted data were quantitatively analyzed using RevMan version 5.4 software to identify the pooled prevalence estimates of the intermediate *Leptospira* spp. [16]. The overall results including heterogeneity represented as *I*^2^ statistic (%) calculated for the included studies were then recorded [17]. Fixed-effect model was used in this study and the overall effect would be considered as statistically significant if the *p*-value was less than 0.05. Sub-group analysis based on United Nations (UN) geo-scheme regions devised by the UN Statistics Division [18] were also carried out to further reduce the heterogeneity or variations between the included studies. Apart from that, the pooled data would highlight the prevalence differences between the regions.

## 3. Results

### 3.1. Literature Search

A total of 469 records were identified from three databases, of which 112 were from ScienceDirect, 286 were from Scopus, 68 were from PubMed, and 3 from the other sources. There were 258 articles removed because they were found to be duplicated and have either the same authors, DOI, or PMID serial number after compiling the results into Microsoft Excel spreadsheet document. Further, the remaining 211 articles were reviewed by title and abstract according to the exclusion and inclusion criteria (Table 2). After excluding 189 records, the remaining 22 articles were subjected for full-text screening according to the eligibility criteria. Then, 15 records were excluded and the remaining seven studies that fit the eligibility criteria were included for meta-analysis. The flowchart of study selection and the reasons for excluding the studies were illustrated in Figure 1.

### 3.2. Characteristics of the Included Articles

There were seven studies selected for meta-analysis, which were published between 2009 to 2020 and the characteristics were detailed in Table 3. The included articles were case-control studies and most of the studies that discovered intermediate *Leptospira* spp. in human samples were from the Asian region and then followed by the American region. There were no eligible studies identified from the European, Oceania, and African regions. The total sample size was 403 while the total samples positive for intermediate and pathogenic *Leptospira* spp. were 225 and 174, respectively. Sera and blood samples collected from humans were utilized for the characterization of the leptospires using several methods such as microscopic agglutination test (MAT), IgM ELISA (enzyme-linked immunosorbent assay), PCR (polymerase chain reaction) assay, partial RNA polymerase β-subunit (rpo-β) gene sequencing, multilocus sequence typing (MLST), 16S rRNA gene sequencing, as well as partial 16S rDNA (rrs) gene sequencing. There were only three out of five intermediate *Leptospira* spp. found from the studies. Five studies found *L. wolffii* [19,20,21,22,23] and one study recorded the presence of both *L. wolffii* and *L. inadai* [10] in their study subjects. Another study reported *L. wolffii* and *L. broomii* [24]. This meta-analysis study showed that *L. wolffii* was the most predominant species (n = 223/225) compared to the other two species, *L. inadai* (n = 1/225) and *L. broomii* (n = 1/225).

### 3.3. Quality Assessment and Risk of Bias

Quality assessment of the selected studies using JBI appraisal checklist for the prevalence study is as shown in Table 4. The mean score of the assessment was 5 out of 9, which ranged from 3 to 9. All of the included studies are deemed to have high risk of bias since none of them were randomized.

### 3.4. Pooled Prevalence of Intermediate Leptospira spp. in Human Samples

The analysis using RevMan software provided the pooled estimates of the input data illustrated as forest plot, as well as upper and lower bounds of 95% confidence interval (CI), *p*-value, and heterogeneity value (*I*^2^ statistic). All the data obtained from the analysis were summarized and the prevalence estimates were bolded in Table 5.

Forest plot in Figure 2 exhibited the overall prevalence estimates on the global prevalence of intermediate *Leptospira* spp. The diagram also showed the first author, year, and the state where the samples were collected. In addition, the proportion outcomes as well as the standard error of the mean (SEM) for the proportion of every study were calculated separately and included manually into RevMan [25]. The statistical analysis revealed that the overall pooled prevalence of intermediate *Leptospira* spp. in humans was 86% (95% CI: 0.85–0.88; *I*^2^ = 99%; *p* < 0.00001).

Subsequently, the studies were categorized by region in accordance with the UN geo-schemes [18]. The pooled prevalence of intermediate *Leptospira* spp. in human samples from the American and Asian regions were 96% (95% CI: 0.94–0.98; *I*^2^ = 96%; *p* < 0.00001) and 17% (95% CI: 0.12–0.23; *I*^2^ = 47%; *p* < 0.00001), respectively (Figure 3 and Figure 4).

## 4. Discussion

Intermediate *Leptospira* spp. have both pathogenic and non-pathogenic serovars, and to date, its role in human pathogenicity remains unclear and is yet to be explored. In this study, the findings from meta-analysis demonstrated that the presence of intermediate *Leptospira* spp. should be considered when making decisions for disease control and prevention, particularly in the regions where leptospirosis is endemic.

To our knowledge, this is the first meta-analysis that summarizes the prevalence of intermediate *Leptospira* spp. in humans worldwide. Meta-analysis of the included studies illuminates that the overall prevalence of the intermediate *Leptospira* spp. is high and significant at 86% (95% CI: 0.85–0.88; *I*^2^ = 99%; *p* < 0.00001). It indicates that these species are indeed contributing to the burden of the disease. Out of two UN regions identified from the included studies, the region of the Americas had the highest prevalence of intermediate *Leptospira* spp., which was at 96% (95% CI: 0.94–0.98; *I*^2^ = 96%; *p* < 0.00001) as compared to the Asian region, which was at 17% (95% CI: 0.12–0.23; *I*^2^ = 47%; *p* < 0.00001) (Table 5). The sub-group analysis suggests that the prevalence of intermediate *Leptospira* spp. in both regions involved was significant. However, the studies found from the European, Oceania, and African regions were excluded because they did not meet the inclusion criteria of this meta-analysis study such that the studies did not specify the intermediate *Leptospira* spp., non-English articles, and some of the studies were irrelevant for this analysis. The data from the included studies also demonstrated that there were only three out of five intermediate *Leptospira* spp. identified, which were *L. wolffii*, *L. inadai*, and *L. broomii*. The other two intermediate species, *L. licerasiae* and *L. fainei* were not found in the selected studies because both species were reported in studies related to non-human samples, which were excluded.

The higher prevalence of intermediate *Leptospira* spp. in the American region than the Asian region indicates that most of the countries in the region of the Americas have more access to health care facilities and better health surveillance system with accurate diagnostic and confirmation methods, which were able to detect and identify the species that infected the patients. This was supported by a report by Schneider et al. (2011) [26], in which they pointed out that the surveillance and control strategies for leptospirosis were developed in many countries in the region of the Americas. Moreover, the improved method in detecting leptospiral DNA has also enabled the identification of intermediate clusters from patients with febrile symptoms in this region [10]. For instance, the utilization of the amplified leptospiral 16S rrs gene and sequencing instead of the common PCR protocols that amplify genes present only in the pathogenic species. Furthermore, there are approximately 10 million people that are affected by natural disasters such as floods (35%) and storms (41%) in the American region every year, and there have been several studies that reported the outbreaks of leptospirosis associated with these events from different countries in Central and South America [27,28]. The results from local studies performed in Central America showed that leptospirosis cases were prevalent among the communities residing in the rural areas, which depend mostly on the animals such as bovine and porcine for their income and for daily protein intake [29]. All these factors might have increased the chances of the communities being exposed to the intermediate *Leptospira* spp. Apart from that, the data also revealed that *L. wolffii*, *L. inadai*, and *L. broomii* were found in the American region (Table 3), which signified that varieties of the intermediate species contributed to the increased burden of leptospirosis in this region.

On the other hand, the prevalence of the intermediate *Leptospira* spp. in the Asian countries, specifically in the South-Eastern Asian and Southern Asian countries, may not be high when compared to the prevalence in the American region, nonetheless it was also statistically significant. The significant presence of the intermediate *Leptospira* spp. in this region, particularly in the Southern Asian countries such as in India and Iran, may be due to the poor access to safe water supplies, poor hygiene, as well as inadequate sanitation. Even though the latest estimates in 2019 showed the improvement in the access to the water supply in India, the water safety and security planning for several districts in India was still lacking and less than 50% of the population has access to safe water supply [30]. Other than that, Zakeri et al. [23] mentioned that 18.5% of the examined cases in northern Iran had collected drinking water from wells and 52% of them had been infected with leptospirosis. The same study also revealed that *L. wolffii* was one of the species isolated from the samples tested. This suggests that unsafe water sources played a role in the transmission of the disease, which may be attributed to the indirect exposure to the intermediate *Leptospira* spp. found in the contaminated water, influencing the incidence of leptospirosis in this region.

In addition, the prevalence of intermediate *Leptospira* spp. in the Asian region was considerably low even though leptospirosis was endemic and causing sporadic outbreaks in most of the South-East Asian developing countries, especially those with humid subtropical and tropical climates such as Malaysia [1,31,32]. This may possibly be due to the diagnostic capabilities of the disease, of which the tools used were less sensitive in detecting the species infecting the patients. Even though culture and microscopic agglutination tests (MAT) are the gold standard methods for laboratory diagnostic testing and the most widely used in this region, they require experts in handling the live pathogens; hence, these methods were offered by only a few hospitals and laboratories in several countries in the Asian region [31]. Additionally, it was said to have little value in predicting the infecting serogroup of the patients since the screening of the serum samples is mostly based on 25 reference serovars, representing only a fraction of over 200 serovars found globally [31,33,34,35,36]. There were several alternative methods to MAT in detecting the acute infection such as enzyme-linked immunosorbent assay (ELISA), IgM dipstick, lateral flow assay, and latex agglutination test, nonetheless, these assays have low sensitivity especially during the acute phase [37,38,39,40]. The accuracies of these techniques are also poor in some areas where leptospirosis is endemic [41,42]. According to Gamage et al. [33], the available laboratory facilities were still poor and inadequate specifically in certain South-East Asian countries, and as reported by WHO in 2009, India, Indonesia, Thailand, and Sri Lanka were the only WHO Member States that have fully or partially implemented laboratory facilities for the diagnosis of leptospirosis [43]. Besides, as most of the South-East Asian countries were the major importers for the agricultural products such as Malaysia, Philippines, and Indonesia [44], the significant prevalence in this region may be contributed to by the occupational factors such as the contact with intermediate *Leptospira* spp. in the contaminated water and soil through farming. This was corroborated with the findings of intermediate *Leptospira* spp. being isolated from the environmental and water samples in several countries in this region [19,45,46].

The data collected from the seven included studies showed that *L. wolffii* was the most predominant species (n = 223/225) as compared to the other intermediate species (*L. inadai*; 1/225; *L. broomii*: 1/225). *L. wolffii* was first isolated from an individual with suspected leptospirosis in Thailand [47], and was also found in all of the included studies, which were in India [19], Argentina [24], Ecuador [10], Iran [20,22,23], and Malaysia [21], which may suggest that *L. wolffii* was the dominant intermediate *Leptospira* circulating in most of the areas. The majority of samples were collected from the patients with acute, febrile illness with other common symptoms for the suspected leptospirosis such as fever, myalgia, chills, rigors, gastrointestinal problems, as well as a more serious symptom like jaundice [10,20,21,22,23]. In one study, several patients with fever, jaundice, hematuria, icteric discoloration with hepatomegaly, as well as weakness on the left side were confirmed to be infected with *L. wolffii* [19]. In other similar studies conducted in Argentina and Malaysia, *L. wolffii* was isolated from patients with fatal cases, particularly respiratory syndrome [21,24]. In addition, as *L. wolffii* was categorized as pathogenic *Leptospira* using nested PCR-RFLP due to the absence of ApoI restriction sites, further sequencing analysis of the samples was required, by which they showed that 26% of the tested DNA belonged to *L. wolffii* [20]. All the tested samples from this study were collected from symptomatic patients manifesting fever with headache, body aches related to jaundice for several days, and headache with myalgia, which all required hospitalization. Furthermore, *L. wolffii* was also isolated from environments and animals such as cattle, rats, pigs, sheep, and dog in several previous reports [10,22,48]. This indicated that *L. wolffii* was prevalent in various environmental and animal reservoirs and that it played a significant role in the transmission cycle of leptospirosis. Therefore, all this evidence suggested that they had the highest pathogenicity compared to other intermediates. *L. broomii*, on the other hand, was identified from one of the human samples in Argentina [24], while *L. inadai* was isolated from one of the human samples in Ecuador [10]. Chiani et al. [24] reported that *L. broomii* was identified from the patient with no signs of severe leptospirosis. This indicates that *L. broomii* likely caused milder disease, which thus explained the smaller number of *L. broomii* being identified from human samples than *L. wolffii*. Meanwhile, *L. inadai* was mainly isolated from animal samples such as cattle, rats, dogs, and pigs [7], suggesting its predominance in the animal reservoirs rather than in humans. It is also noteworthy that one of the included studies found that 96% of leptospiral DNA isolated from human serum belonged to the intermediate species including *L. wolffii* and *L. inadai*, rather than pathogenic cluster strains [10].

Despite the increasing studies concerning leptospirosis disease in the African region, there were no eligible studies that reported the intermediate *Leptospira* spp. in this region, which may be due to the inadequate sampling and poor access to the diagnostic facilities, leading to the under-reporting of leptospirosis. Hence, there is insufficient information regarding the species that infected the patients from the African region [49]. Besides, there were also no eligible studies that reported the presence of intermediate *Leptospira* spp. in the Oceania region, albeit the previous systematic review had revealed that the incidence of leptospirosis was notably high (150.68 cases per 100,000 per year), especially in the temperate parts of the region such as Australia and New Zealand [50,51]. However, there were too little information and investigations regarding the specific species that infected the patients from this region. Other than that, there were also no eligible studies that reported the presence of intermediate *Leptospira* spp. in the European region. This was because leptospirosis was not a common disease with 0.2 confirmed cases per 100,000 population, which is considered a low rate in European countries in comparison with other regions [52].

There were several limitations of this study, which include the possibility of missing some of the relevant studies during the literature search procedure. Also, the information from the other countries and regions was lacking. This prevented us from making a thorough analysis of the prevalence data needed for species-wise stratification, reducing the accuracy of the obtained data. This also limited our study from analyzing the pathogenicity status of the intermediate *Leptospira* spp. Furthermore, the high risk of bias of all the studies included and high *I*^2^ statistic for the overall prevalence (99%) might influence the accuracy of prevalence estimates in this analysis. Differences in the design, complexity, geographical regions, environments, and study settings of the included studies contributed to the high heterogeneity in this analysis. However, the chances of getting correct estimates are higher using a random-effect model with increasing heterogeneity, than using fixed-effect model [53]. Thus, optimizing the effect model based on the heterogeneity instead of outright rejecting the result due to the prominent heterogeneity is preferred [54]. Other than that, publication bias assessment using a funnel plot was not carried out since there were only seven included studies in this study. Higgins et al. [17] stated that the power of the tests would be too low to distinguish the chance from the real asymmetry if the studies are fewer than 10. Additionally, Debray et al. [55] also reported that the power for the tests of the funnel plot asymmetry usually remained less than 50% even when there were ≥ 50 studies available for meta-analysis. Therefore, it is best to use a funnel plot only when there is a minimum of 10 studies. Lastly, based upon the advanced literature survey, we found only a few articles that addressed the presence of intermediate *Leptospira* spp. and identified the exact type of species in human samples.

## 5. Conclusions

In conclusion, this is the first meta-analysis on the prevalence of intermediate *Leptospira* spp. from human samples worldwide. The overall prevalence estimate of intermediate *Leptospira* spp. was high and statistically significant, and the pooled prevalence estimates based on the UN regions showed the highest prevalence of intermediate *Leptospira* spp. in the American region followed by the Asian region. The data from the included studies also demonstrated that *L. wolffii* was the most predominant species found in the human samples as compared to *L. inadai* and *L. broomii*. All the findings suggest that intermediate *Leptospira* spp. played an important role in the transmission of human leptospirosis. This calls for more investigations using molecular analysis, as it would give accurate species identification, which can be used to break the chain of leptospirosis and reduce the disease burden. Also, further studies on the effect of the species on the clinical outcome of the patients are required to gain better understanding on the pathogenicity status and capacity of intermediate *Leptospira* spp.

## Figures and Tables

**Figure 1 pathogens-10-00943-f001:**
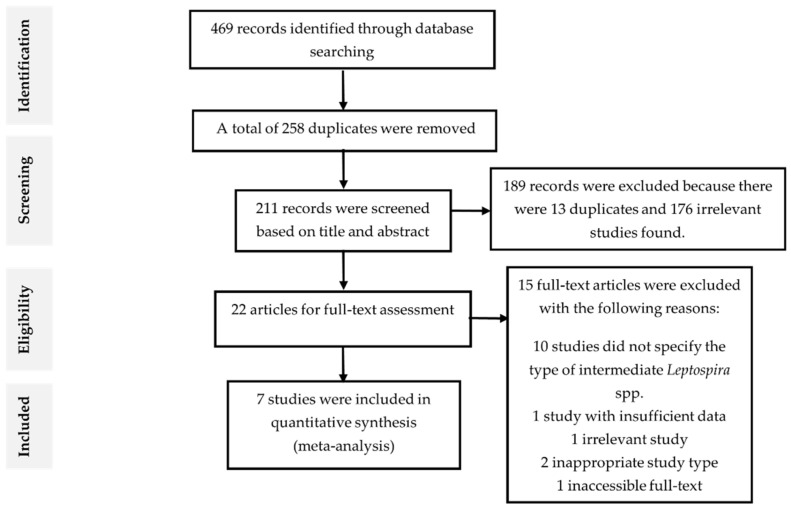
Flow diagram of literature search and selection.

**Figure 2 pathogens-10-00943-f002:**
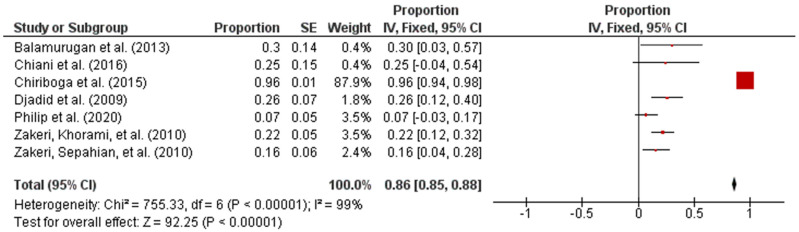
Forest plot of overall prevalence of intermediate *Leptospira* spp. in human samples.

**Figure 3 pathogens-10-00943-f003:**
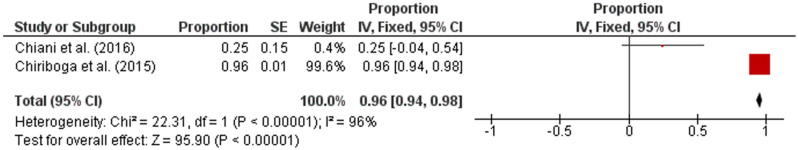
Forest plot of pooled prevalence of intermediate *Leptospira* spp. in human samples from the American region.

**Figure 4 pathogens-10-00943-f004:**
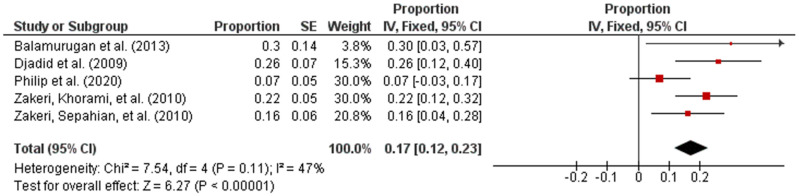
Forest plot of pooled prevalence of intermediate *Leptospira* spp. in human samples from the Asian region.

**Table 1 pathogens-10-00943-t001:** Search terms and keywords for the literature search.

	PubMed	Scopus	ScienceDirect
*Leptospira broomii*	(prevalence OR epidemiology) AND leptospir * AND (* leptospira broomii” OR “l.broomii” OR “intermediate leptospir *) AND (human * OR patient *)	TITLE-ABS-KEY (prevalence OR epidemiology) AND leptospir * AND (“leptospira broomii” OR “l.broomii” OR “intermediate leptospir *”) AND (human * OR patient *)	(prevalence OR epidemiology) AND leptospirosis~) AND (“leptospira broomii” OR “l.broomii” OR “intermediate leptospira”) AND (human~ OR patient~)
*Leptospira fainei*	(prevalence OR epidemiology) AND leptospir * AND (* leptospira fainei” OR “l.fainei” OR “intermediate leptospir *) AND (human * OR patient *)	TITLE-ABS-KEY (prevalence OR epidemiology) AND leptospir * AND (“leptospira fainei” OR “l.fainei” OR “intermediate leptospir *”) AND (human * OR patient *)	(prevalence OR epidemiology) AND leptospirosis~) AND (“leptospira fainei” OR “l.fainei” OR “intermediate leptospira”) AND (human~ OR patient~)
*Leptospira wolffii*	(prevalence OR epidemiology) AND leptospir * AND (* leptospira wolffii” OR “l.wolffii” OR “intermediate leptospir *) AND (human * OR patient *)	TITLE-ABS-KEY (prevalence OR epidemiology) AND leptospir * AND (“leptospira wolffii” OR “l.wolffii” OR “intermediate leptospir *”) AND (human * OR patient *)	(prevalence OR epidemiology) AND leptospirosis~) AND (“leptospira wolffii” OR “l.wolffii” OR “intermediate leptospira”) AND (human~ OR patient~)
*Leptospira licerasiae*	(prevalence OR epidemiology) AND leptospir * AND (* leptospira licerasiae” OR “l. licerasiae” OR “intermediate leptospir *) AND (human * OR patient *)	TITLE-ABS-KEY (prevalence OR epidemiology) AND leptospir * AND (“leptospira licerasiae” OR “l. licerasiae” OR “intermediate leptospir *”) AND (human * OR patient *)	(prevalence OR epidemiology) AND leptospirosis~) AND (“leptospira licerasiae” OR “l. licerasiae” OR “intermediate leptospira”) AND (human~ OR patient~)
*Leptospira inadai*	(prevalence OR epidemiology) AND leptospir * AND (* leptospira inadai” OR “l. inadai” OR “intermediate leptospir *) AND (human * OR patient *)	TITLE-ABS-KEY (prevalence OR epidemiology) AND leptospir * AND (“leptospira inadai” OR “l. inadai” OR “intermediate leptospir *”) AND (human * OR patient *)	(prevalence OR epidemiology) AND leptospirosis~) AND (“leptospira inadai” OR “l.inadai” OR “intermediate leptospira”) AND (human~ OR patient~)

Note: * or ~ are the truncation symbols added to the start or end of the search term to identify the articles with every word that could have various endings and spellings.

**Table 2 pathogens-10-00943-t002:** Eligibility criteria for study selection.

Picos Element	Inclusion Criteria	Exclusion Criteria
Population	Any individual suspected with leptospirosis infection (or co-infected with the other diseases)	Not leptospirosis infection
	Any age and gender	Irrelevant study
		Studies that do not report the origin of the samples or patients
Intervention	Studies specified the species of the intermediate *Leptospira* spp.	Studies that detect the presence only but do not identify the type of species
	Studies specified any diagnostic or confirmation methods used	Studies that do not report the method used
		Studies that report the presence of leptospires in animals and environmental samples only
Comparator	Studies reported the number of samples positive for pathogenic *Leptospira* spp.	Insufficient or unclear data
	Studies reported the number of total samples	
Outcome	Studies reported the number of samples positive for intermediate *Leptospira* spp.	
	Studies reported the number of total samples	
Study Design	Research articles of any countries and any publication year	Inaccessible full-text article
		Letter to editor
		Duplicate publications
		Foreign language (other than English)
		Secondary research (review papers, systematic review, and meta-analyses)
		Letter to editor

**Table 3 pathogens-10-00943-t003:** Characteristics of the included studies.

Species of the Intermediate (No. of Samples)	State	Country	Regions	No. of Intermediate Positive Samples/Total Number of Confirmed Cases (% Prevalence)	No. of Pathogenic Positive Samples/Total Number of Confirmed Cases (% Prevalence)	Methodology	Type of Study Design	First Author and Year of Publication
*L. wolffii* (3)	Karnataka	India	Asia (Southern Asia)	3/10 (30.0%)	7/10 (70.0%)	PCR assay, partial RNA polymerase β-subunit (rpo-β) gene sequencing	Case-control	Balamurugan et al., 2013 [19]
*L. wolffii* (1) and *L. broomii* (1)	Santa Fe and Buenos Aires	Argentina	Americas (South America)	2/8 (25.0%)	6/8 (75.0%)	16S rRNA gene sequencing and MLST	Case-control	Chiani et al., 2016 [24]
*L. wolffii* (129)	Esmeraldas	Ecuador	Americas (South America)	129/132 (97.7%)	3/132 (2.3%)	IgM ELISA, real-time PCR, rrs sequencing	Case-control	Chiriboga et al., 2015 [10]
*L. wolffii* (24) and *L. inadai* (1)	Portoviejo			25/25 (100.0%)	0/25 (0.0%)			
*L. wolffii* (28)	Guayaquil			28/32 (87.5%)	4/32 (12.5%)			
*L. wolffii* (11)	Guilan	Iran	Asia (Southern Asia)	11/42 (26.2%)	27/42 (64/3%)	PCR-RFLP assay, nested PCR−16S rRNA gene sequencing	Case-control	Djadid et al., 2009 [20]
*L. wolffii* (2)	Selangor	Malaysia	Asia (South-Eastern Asia)	2/28 (7.1%)	26/28 (92.9)	MAT, PCR, partial 16S rDNA (rrs) gene sequencing	Case-control	Philip et al., 2020 [21]
*L. wolffii* (18)	Mazandaran, Guilan, Ardebil, and Tehran	Iran	Asia (Southern Asia)	18/82 (21.9%)	64/82 (78.0%)	Nested PCR/RFLP analysis, 16S rRNA gene sequencing	Case-control	Zakeri, Khorami, et al., 2010 [22]
*L. wolffii* (7)	Mazandaran	Iran	Asia (Southern Asia)	7/44 (15.9%)	37/44 (84.1%)	Nested PCR/RFLP analysis, 16S rRNA gene sequencing	Case-control	Zakeri, Sepahian, et al., 2010 [23]

**Table 4 pathogens-10-00943-t004:** Quality assessment of the included studies.

Author and Year	Q1	Q2	Q3	Q4	Q5	Q6	Q7	Q8	Q9	Total
Balamurugan et al., 2013 [19]	U	1	0	1	0	1	U	0	U	3
Chiani et al., 2016 [24]	U	1	U	1	0	1	1	0	1	5
Chiriboga et al., 2015 [10]	1	1	1	0	0	1	1	0	U	5
Djadid et al., 2009 [20]	U	1	U	0	0	1	1	0	1	4
Philip et al., 2020 [21]	1	1	0	1	U	1	1	1	U	6
Zakeri, Khorami, et al., 2010 [22]	0	1	U	1	U	1	1	0	U	4
Zakeri, Sepahian, et al., 2010 [23]	1	1	1	1	1	1	1	1	1	9

Q1: Was the sample frame appropriate to address the target population?; Q2: Were study participants sampled in an appropriate way?; Q3: Was the sample size adequate?; Q4: Were the study subjects and the setting described in detail?; Q5: Was the data analysis conducted with sufficient coverage of the identified sample?; Q6: Were valid methods used for the identification of the condition?; Q7: Was the condition measured in a standard, reliable way for all participants?; Q8: Was there appropriate statistical analysis?; Q9: Was the response rate adequate, and if not, was the low response rate managed appropriately?; 0: No; 1: Yes; U: Unclear.

**Table 5 pathogens-10-00943-t005:** Meta-analysis of the prevalence of intermediate *Leptospira* spp. in humans.

Study	Number of Articles	Heterogeneity (%)	Effect Size	95% Confidence Interval (CI)		
			Prevalence (%)	*p*-Value	Lower Value	Upper Value
Prevalence of intermediate *Leptospira* spp. in humans	7	99	**86**	*p* < 0.00001	0.85	0.88
***Region***-***Wise***						
UN American region	2	96	**96**	*p* < 0.00001	0.94	0.98
UN Asian region	5	47	**17**	*p* < 0.00001	0.12	0.23

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
