# Peer review of "Regional Prevalence of Intermediate Leptospira spp. in Humans: A Meta-Analysis"

_pathogens, 2021, doi:10.3390/pathogens10080943_

Round 1
Reviewer 1 Report
TITLE : The title is not consistent with the content. It's not an assessment of global intermediate Leptospira spp. The "Global" term needs to be changed, because it is not consistent with the fact that it's data mainly from Asia and South America.
But my objection goes to the title for the term "global". This term indicates the whole earth, but as the authors themselves say it does not include data on Europe, Oceania and Africa regions. If possible I would like that in paragraph 3.2, in addition to indicating the methods used for identifying cases of disease, they also indicate the medical data of at least the most significant cases. This would give greater importance to their study especially when intermediate Leptospira spp. were present.Discussion : you should give more information about the presence or not of medical data reported in the work (presence of any symptoms, pathology) as you did from the 338 line to 348 line). This report is only the data on prevalence as statistical data, but from the medical point of view you give little information. I think the prevalence of intermediate Leptospira spp. is an end in itself if you don't indicate beyond the part of the laboratory analysis what's reported in these work (as an impact on human health).
Reviewer 2 Report
Thanks for giving me an opportunity to review this paper. This paper is well written and the overall presentation is good. However, there are several issues that need to fix.
- Authors should mention the study design of included studies; it is unclear whether they are cohort, case-control or cross-sectional studies.
- Section 2.4: Authors mention that studies bias was assessed in accordance with the modified Critical Appraisal Checklist recommended by the Joanna Briggs Institute (JBI)" . However, there are several assessment criteria recommended by JBI, it is unclear which criteria they have used. Another question is, why authors did not consider Cochrane recommended criteria?
- Section 2.4, line 164-165: "If the ?2 statistic was low or less than 50%, it indicates that heterogeneity was not significant and thus fixed-effect model would be used...". Can you provide a valid reason? please cite the paper here.
- Section 3.3, line 212-213: "The study was considered as a low-quality study if the total score was less than 5. Meanwhile if the study scored equal or more than 5, the study was considered as a high-quality study". please give the previous evidence on how have you categorized it.
- Table 3: please provide the study design.
- Table 4: Please mention the quality assessment for what type of study.
- study heterogeneity is high (99%) how would the authors explain it?
- Discussion, Line 384-385: "publication bias assessment using funnel plot could not be carried out since there were only seven included studies in this study which did not reach the minimum studies required for the test which was 10". It is not right, you can draw a funnel plot using only 4 studies. Please provide evidence.
Round 2
Reviewer 1 Report
You did not change the title as I asked you. I think that the term "Global" remains wrong.
From Line 311 to 318 : Where is the additional information about the intermediate Leptospira spp. added in
the discussion ?
Reviewer 2 Report
Thanks for your revised version. It is well shaped now but there are some issues that need to fix before considering it for publication.
- Figure 2,3,4, please put only author first/last name with year. It should be Balamurugan 2013, not Balamurugan 2013 Karnataka. Please correct all those figures.
- I found some discrepancies in Proportion IV random, 95%CI in those figures from original articles. Please correct all these discrepancies. It should be the same as the original articles.
Round 3
Reviewer 1 Report
In this revision of the manuscript you have made the changes that I have requested.
Author Response
Thank you for the previous constructive comments and for reviewing our manuscript.
Reviewer 2 Report
Thanks for your revised version. It is well shaped now but far away to be perfect.
First, please check again prevalence discrepancy and calculate again.
Second, you do not need to put regional names in the figure to dedifferentiate the data from different regions of the country. You can just write Chiriboga et al. (2015) and describe it figure caption. Otherwise, it looks weird and people will be confused. Please correct the figures and describe everything in the caption.
Third, Chiriboga et al. (2015) portovijo: Proportion IV is 1 (1.00-1.00). That mean prevalence is 100%. I don't think it is right, please check it again
